# Allosteric Binding Sites of Aβ Peptides on the Acetylcholine Synthesizing Enzyme ChAT as Deduced by In Silico Molecular Modeling

**DOI:** 10.3390/ijms23116073

**Published:** 2022-05-28

**Authors:** Anurag TK Baidya, Amit Kumar, Rajnish Kumar, Taher Darreh-Shori

**Affiliations:** 1Department of Pharmaceutical Engineering & Technology, Indian Institute of Technology (B.H.U.), Varanasi 221005, Uttar Pradesh, India; anuragtkbaidya.rs.phe20@itbhu.ac.in; 2Division of Clinical Geriatric, Center for Alzheimer Research, Department of Neurobiology, Care Sciences and Society, Karolinska Institutet, NEO, 141 52 Stockholm, Sweden; amit.kumar@ki.se

**Keywords:** β-amyloid, choline acetyltransferase, cholinergic system, Alzheimer’s disease, in silico modeling

## Abstract

The native function of amyloid-β (Aβ) peptides is still unexplored. However, several recent reports suggest a prominent role of Aβ peptides in acetylcholine homeostasis. To clarify this role of Aβ, we have reported that Aβ peptides at physiological concentrations can directly enhance the catalytic efficiency of the key cholinergic enzyme, choline acetyltransferase (ChAT), via an allosteric interaction. In the current study, we further aimed to elucidate the underlying ChAT-Aβ interaction mechanism using in silico molecular docking and dynamics analysis. Docking analysis suggested two most probable binding clusters on ChAT for Aβ_40_ and three for Aβ_42_. Most importantly, the docking results were challenged with molecular dynamic studies of 100 ns long simulation in triplicates (100 ns × 3 = 300 ns) and were analyzed for RMSD, RMSF, RoG, H-bond number and distance, SASA, and secondary structure assessment performed together with principal component analysis and the free-energy landscape diagram, which indicated that the ChAT-Aβ complex system was stable throughout the simulation time period with no abrupt motion during the evolution of the simulation across the triplicates, which also validated the robustness of the simulation study. Finally, the free-energy landscape analysis confirmed the docking results and demonstrated that the ChAT-Aβ complexes were energetically stable despite the unstructured nature of C- and N-terminals in Aβ peptides. Overall, this study supports the reported in vitro findings that Aβ peptides, particularly Aβ_42_, act as endogenous ChAT-Potentiating-Ligand (CPL), and thereby supports the hypothesis that one of the native biological functions of Aβ peptides is the regulation of acetylcholine homeostasis.

## 1. Introduction

Alzheimer’s disease (AD) is the most prevalent form of dementia affecting millions of people worldwide. It is characterized by the presence of pathological markers such as senile plaques and neurofibrillary tangles in the brain of AD patients. Senile plaques are mainly composed of aggregated Amyloid-β peptide (Aβ; product of amyloid precursor protein (APP) cleavage) [1]. Aβ peptides can range from 36 to 43 amino acids depending on the enzyme cleavage sites, but the most common pathological forms are Aβ_40_ and Aβ_42_ [1,2]. Aβ_42_ is considered more toxic and highly prone to aggregation and accumulation as compared to Aβ_40_. Another pathological feature, prominently found in AD brains, is cholinergic dysfunction that has been linked to cognitive decline and memory impairment in AD patients [3]. Indeed, loss of basal nuclei cholinergic neurons is shown to be directly associated with cognitive impairment in AD [4,5]. However, the exact cause of cholinergic deficit and how it is related to other pathological changes in AD is still puzzling researchers around the world.

Cholinergic neurons, defined by the expression of the acetylcholine (ACh) synthesizing enzyme choline acetyltransferase (ChAT; EC 2.3.1.6), are projected from certain nuclei in the basal forebrain throughout the brain [6] and regulate key cognitive functions such as attention, memory, and learning. It has been shown that the cholinergic signaling machinery, including high-affinity choline transporter (hChT) and ChAT, is highly susceptible to Aβ peptides and its aggregated forms [4,7,8,9].

The innate physiological function of Aβ peptides is still unknown and the focus of the research community has been on the toxicity of these peptides and their aggregates. However, studies have shown that Aβ can act as a ligand of RAGE (receptor for advanced glycation end products) and regulate key inflammatory signaling pathways [10]. In addition, physiological concentrations of Aβ_42_ peptide can activate the MAPK (mitogen-activated protein kinase) cascade in the hippocampus via the α7 nicotinic acetylcholine receptor [11]. In mammals, MAPK cascade is involved in learning and memory formation [12]. Most importantly, recent studies have suggested a more prominent role of Aβ peptides in synaptic and extrasynaptic ACh homeostasis. We have previously shown that Aβ peptides interact with acetyl- and butyryl-cholinesterase (AChE and BChE) in an apolipoprotein facilitated manner, and thereby can modulate cholinergic signaling by forming highly stable and ultra-active soluble ACh-degrading complexes called BAβACs [13].

Other studies have shown that short- and long-term exposure to Aβ peptides can lead to a reduced ACh synthesis and release from neuronal cell cultures as well as neurodegeneration and cognitive impairment in animal models [14,15,16,17,18,19,20,21,22,23,24,25]. However, it is not clear whether these effects are caused by a direct or an indirect mode of action of Aβ peptides on the activity of the core cholinergic enzyme ChAT. In contrast, we have shown that Aβ peptides, in particular Aβ_42_, directly interact and allosterically enhance rather than reduce the catalytic rate of ChAT at physiological concentrations representative of Aβ levels in the CSF of AD patients and healthy controls (Figure 1) [26].

Given that cholinergic deficit is one of the key features in major dementia disorders, the findings that Aβ peptides can allosterically increase biosynthesis of ACh opens a new avenue for the development of a new class of cholinergic enhancing therapeutic strategy based on improving ACh biosynthesis that we are calling ChAT-Potentiating-Ligands (CPLs), as opposed to the cholinesterase inhibitors approach based on inhibiting the degradation of ACh by cholinesterases. In the primary paper of our study, we have reported that Aβ_40_ exhibited CPL activity with an EC_50_ value of 800 pg/mL (or 185 pM) while Aβ_42_ peptides showed a 10-fold higher CPL potency as deduced by an EC_50_ value of 68 pg/mL (or 15 pM) [26]. These EC_50_ values are fully in the range of the expected concentration of Aβ peptides in the human brain, as deduced by the known concentration of Aβ peptides in human cerebrospinal fluid. Thereby, it is imperative to determine the molecular fingerprint of the interaction between ChAT and the Aβ peptides as the only known endogenous CPL.

Therefore, the current study had two main objectives. Firstly, we aimed to shed further light on the physiological role of Aβ peptides in ACh regulation, and secondly, to investigate and elucidate the molecular fingerprint of the binding mechanism of Aβ peptides on ChAT governing the CPL activity of the Aβ peptides, using advanced in silico molecular docking and molecular dynamics studies. We report here the results of both the docking and 100 ns molecular simulation analyses in terms of the molecular dynamic parameters, such as root means square deviation (RMSD), radius of gyration (Rg), root mean structure fluctuation (RMSF) and H-bond functionality values. We show that these analyses indicate that Aβ peptides can physically interact with ChAT protein, forming highly stable ChAT-Aβ complexes as deduced by a free-energy landscape analysis. In addition, we provide details on the stability and robustness of the ChAT-Aβ peptides simulated system.

## 2. Results

Amyloid-β and its exact native function are still questions of debate among researchers worldwide. In our recently published paper, we have shown that Aβ peptides can upregulate the activity of the core-cholinergic enzyme ChAT by direct interaction at physiological concentration ranges and this enhanced catalytic efficiency was persistent even at the physiological ratio of Aβ_40_:Aβ_42_ mixture (Figure 1; adapted and reproduced from Kumar et al. [26], under the terms of the CC BY license). Here, molecular docking studies were performed using a Cluspro protein-protein docking web server to elucidate the possible interaction mechanism of Aβ peptides with ChAT. The final protein-protein docking cluster analysis identified 29 clusters for ChAT-Aβ_40_ and ChAT-Aβ_42_ complexes. The most probable binding cluster for ChAT-Aβ_40_ complex Cluster-0 with a maximum of 88 members and the lowest energy (−989.5 kcal/mol). The corresponding was Cluster-0 for ChAT-Aβ_42_ complex had the maximum members of 134 and the lowest energy of −999.6 kcal/mol (Table 1).

The predicted models were well justified based on the scores, as it implies the linear combinations of several energy terms, such as the attractive and repulsive van der Walls energy describing the complementary shape, and one or more terms denoting the electrostatic binding energy and cluster sizes. The 3D cartoon representations for the top three clusters of ChAT-Aβ_40_ are given in (Figure 2B–D). The putative entrances of the catalytic tunnel on ChAT are indicated by a *red circle* (for choline/ACh entrance site) and *yellow circle* (for ACoA/-CoA entrance site; Figure 2). Intriguingly, in the cluster-1, Aβ_40_ seemed to completely cover the mouth of the choline/ACh entrance site of ChAT’s catalytic tunnel (*red circle* in Figure 2C), whereas the most probable cluster-0 indicated that Aβ_40_ nearly touched the ACoA/-CoA entrance site on ChAT (*yellow circle* in Figure 2B) as well as the cluster-2 indicating that the Aβ_40_ covers the mouth of ACoA/-CoA entrance site on ChAT (*yellow circle* in Figure 2D). Thus, these binding cluster analyses intuitively suggest that the Aβ_40_ binding at cluster-1, but not the cluster-0 or -2, could block the influx of the substrate, choline or ACh, and thereby the access of the enzyme to these substrates. Importantly, this could explain why in some of our in vitro experiments, the Aβ_40_ either inhibited or induced minimal activation of ChAT (Figure 1C) [26].

For ChAT-Aβ_42_ complex, the top three clusters are presented as a 3D cartoon in (Figure 2F–H), illustrating the most probable binding sites of Aβ_42_ on ChAT. Again, the position of the choline/ACh entrance site of the catalytic tunnel is highlighted by a *red circle* (in Figure 2E).

Surprisingly, like Aβ_40_, Aβ_42_ appeared to completely overlap the catalytic tunnel, but in this case it was in the most probable cluster-0 (Figure 2F), and not in the cluster-1 like Aβ_40_. Nonetheless, there are two additional binding sites for Aβ_42_ that are in close vicinity of the mouth of the ACoA/-CoA entrance site on the catalytic tunnel, namely the clusters -1 and -2 (*yellow circle*; Figure 2G,H). These sites might best explain the improved catalytic efficiency of ChAT by Aβ_42_ peptides (compare Figure 1A,B). In addition, we have compiled the simultaneous possible bindings for Aβ_42_ and Aβ_40_ as depicted in (Figure 2I). For instance, binding of Aβ_42_ at cluster-1 (Figure 2G) will most likely exclude binding of Aβ_40_ at cluster-0 (Figure 2B) or vice versa. In contrast, it is also possible that two Aβ bind simultaneously on a ChAT molecule, e.g., conditions C-H or F-H in (Figure 2I).

Next, the predictions of the binding mode of Aβ peptides to ChAT by the docking analyses were challenged with advanced molecular dynamic (MD) simulation analysis in order to realize the conformational changes and stability during the course of interaction. The time required for the biological phenomenon to occur is considered as the optimum time for simulation. Based on the resources available, the calculations were tuned and, keeping our objective in mind to fulfil the criteria that the RMSD stabilizes over the period of time, were indicated by a stable line that concludes that the system has reached convergence and is comparatively stable. To explore the global minima in the energy landscape, the MD trajectories were run and analyzed under physiological environment for a 100 ns-long simulation time period in triplicates. The results of the molecular dynamic analysis of the first primary run are shown in (Figure 3) as a measure of the stability and quality of the simulated system. The RMSD plot suggested that the ChAT protein remained stable from 0.1–0.28 nm, as shown in (Figure 3A). The averages of the triplicates were plotted, as represented in Appendix A. The results are well within the limits with low uncertainties, which are denoted as the ‘Y’ bar error (standard deviation) obtained from three independent 100 ns simulations. The RMSD value of Aβ peptides for the first run (Figure 3B) seems to be in the ranges of 0.3–1.73 nm, which indicates their stability in the complex system throughout the simulation time. Likewise, the averages obtained from the triplicate runs were plotted, as shown in Appendix A. The standard deviation is larger compared to the protein structure, which is due to the flexibility of the peptide as it interacts with the protein surface throughout the course of simulation. The radius of gyration (RoG) of the complex was also determined, as shown in (Figure 3C), to further understand the compactness and overall dimensions of the protein structure. The RoG values for the first run were in line with the RMSD values as well as stable and relatively low ranging between 2.5 and 2.65 nm for the ChAT-Aβ complex. This indicated that the formed complex retained a compact folded conformation throughout the simulation time. The individual RoG plots for all three runs of the ChAT-Aβ complex clusters are shown in Appendix A. The results from the triplicates indicate that the RoG values remained stable with relatively low variation that were in an acceptable range.

Furthermore, to understand the dynamic behaviors of the amino acid residues and flexibility of the protein structure during the interaction period of the simulation, we recorded and analyzed the root mean square fluctuation (RMSF) as a measure of individual amino acid fluctuation throughout the course of simulation. The magnitude of individual amino acid residual flexibility is represented by the corresponding height of the peaks. RMSF plots of ChAT protein for the first run indicate no major fluctuation in the ChAT core structure (Figure 3D), with a value of 0.05–0.4 nm for the amino acid residues. However, the C- and N-terminals of the ChAT protein exhibited higher structural flexibility with values ranging up to 0.7 nm. This is expected since they are naturally exposed to solvent surface. The average RMSF values of the ChAT protein, along with the standard deviation for the three runs are shown in Appendix A, which shows low fluctuations. Likewise, the RMSF value of the Aβ peptides (Figure 3E) was observed between the acceptable limits of 0.1–1.2 nm indicating the relative stability of the Aβ peptides in the system. The C- and N-terminal of Aβ showed higher RMSF value, indicating the higher flexibility of the regions due to the unstructured nature, as reported based on NMR structural studies [27,28]. The average RMSF values of the Aβ peptides, along with the standard deviation for the three runs is shown in Appendix A. The larger variation most likely reflects the higher flexibility of the C- and N-terminal of Aβ peptides. Overall, the RMSF suggested that Aβ was tightly bound with the ChAT protein, despite these C- and N-terminal flexibilities and the overall ChAT-Aβ complexes remained stable for the whole simulated time.

Hydrogen bonding plays a central role during the interactions, formation and stabilization of protein complexes. The typical distance criterion for a hydrogen bond donor and acceptor atoms is ≤3.5 Å (0.35 nm) with an angle of 180° ± 30° between hydrogen bond donor and acceptor. Therefore, we also calculated the average number of hydrogen bonds formed during ChAT-Aβ peptide interactions throughout the 100 ns simulation with a cut-off value of 0.35 nm, as shown for the first run in Figure 4A. The individual landscapes for the hydrogen bonds formed between ChAT-Aβ peptides during all the three runs is shown in Appendix A. The results were consistent with respect to each other, which indicates that a stable number of hydrogen bonds were maintained throughout the simulation period. The average hydrogen bond distance was also calculated, which was found to be about 0.3 nm, which is within the expected range (Figure 4B). The results from all the triplicates were very similar (Appendix A), with an average hydrogen bond distance of about 0.3 nm.

Moreover, we performed a solvent accessible surface area (SASA) analysis, which measures the surface area of the biomolecule accessible by the solvent (biofluids), and helps in predicting the stability of the hydrophobic core of the protein. These hydrophobic contacts are crucial indicators of the compactness of the tertiary structures of a protein [29]. We found that the SASA profile of each ChAT-Aβ cluster complex is consistent with its radius of gyration as there was a slight decrease in SASA. This in turn indicates a contracted conformation of the complex, as a higher SASA profile reflects a relative expansion of the protein surface area and would have resulted in a more fluctuating radius of gyration. The values ranged from 250–275 nm^2^ for ChAT proteins (Figure 4C) and 39–49 nm^2^ for Aβ peptides (Figure 4D) for the first run. The averages from the triplicates along with their standard deviation is shown in Appendix A. The results were consistent with reasonably low variations in the reproduction of the SASA profiles. Furthermore, SASA for the ChAT protein seemed to be reduced at the end of the simulation (Figure 4C), indicating that the Aβ peptides are interacting with the ChAT protein and are causing changes in the structure and surface residues of the ChAT protein in such a way that it further shields its hydrophobic core accessible to the solvent. Whilst for the Aβ peptides, SASA showed an increase towards the end of the simulation with constant changes due to the interaction with ChAT protein.

To unravel the connection between the internal motion and the secondary structures, such as the coil, beta sheets, beta-bridge, bend, turn, and helices throughout the simulation time, a Dictionary of Secondary Structure of Protein (DSSP) analysis was instigated, which identifies protein sheets and helix assignments solely on the basis of backbone–backbone hydrogen bonds. We performed TESSE (time-evolution of the secondary structural elements) analyses for ChAT and Aβ peptides, which are visually summarized in Appendix A, respectively. The analyses in (Appendix A) further reveal that the ChAT protein maintained its secondary structure throughout the simulation period without much conformational change, since all the amino acid residues maintained their structure during the simulation period.

Moreover, we observed that the ChAT protein retained a greater number of alpha helices, which is about 40% of the amino acids remained in this state, as compared to coils, beta sheet, turn, bend, 5-helices, and 3-helices. This observation is based on data presented in Table 2 and Table 3, which show the percentage of secondary structure distribution in the trajectory clusters for Aβ_40_ and Aβ_42_ on ChAT in each cluster system, respectively. The occurrence and maintenance of alpha helices is the most stable dominant structural element. This was a common feature of protein secondary structure in the ChAT protein throughout the simulation period, reflecting its compactness, energetically most favorable conformations, and slow steady change throughout the simulation. On the other hand, the Aβ peptides seemed to be constantly changing over the simulation time period, as is evident by containing a greater number of turns and random coils (Appendix A), which could reflect an interplay with the ChAT catalytic mouth of Choline/ACh entrance site.

We did a Principal Component Analysis (PCA), also known as essential dynamics based on the principles of covariance matrix, to identify the most important conformational degrees of freedom of the simulation system. It helps in shortening the large dimensions of the data set to the major principal components that represents the considerable variations that can sufficiently explain the overall motions of the protein. The eigenvalues for the complex are shown in (Figure 5), where it is clearly evident that the first four components explain the most variation in the data which is 53.58, 78.71, and 52.70 for cluster-0, -1, and -2 of Aβ_40_ peptides, respectively, and in case of Aβ_42_ peptides it is about 72.85, 49.25, and 52.43 for the cluster-0, -1, and -2, respectively. The plot shows that the eigenvalues after the fourth principal component starts to make a straight horizontal line. Thus, we have selected the first two principal components to plot our essential dynamics plot.

The plotted 2D graph for the first two principal components (Figure 6), represents the various conformations taken by the protein in the course of simulated period. The changes in the ChAT-Aβ complex system indicates that Aβ peptides affected the flexibility of the ChAT protein. ChAT-Aβ_40_ cluster-0 and -2 and ChAT-Aβ_42_ cluster-1 and -2 complexes demonstrated concentrated conformational sampling, indicating a stable state conformation occupied by the overall system. While the ChAT-Aβ_40_ cluster-1 and the ChAT-Aβ_42_ cluster-0 display a widespread distribution of the conformational space, this can be due to larger conformational changes in the protein structure, thus indicating that a wide range of conformational states are occupied by the complex over the simulation time.

Free-energy landscape (FEL) is a representation of possible conformations taken by a protein in molecular dynamics simulation along with the Gibbs free energy. FEL represents two variables that reflect specific properties of the system and measure conformational variability. To visualize the energy minima landscape of bound ChAT-Aβ complex, we studied the free-energy landscape (FEL) against radius of gyration (RoG) and root-mean-square deviation (RMSD) as the two reaction coordinates. This revealed the changes in the Gibbs free energy (ΔG) values between 0 and 9.950 kJ/mol for Aβ_40_ (Figure 7) and from 0 to 10 kJ/mol for Aβ_42_ (Figure 8) complexes.

The shape and size of the minimal energy area (shown in blue) indicate the stability of the ChAT-Aβ complex system. Smaller and more centralized blue areas represent the complex within the cluster with the highest stability. The narrow funnel formed as seen in the 3D projections show the dynamic of the change in conformation with the time of simulation required to attain a native structure with least energy. Overall, the 3D plots illustrate that the overall complexes formed a single funnel, which together with the 2D contour plot clearly indicates the complexes having one local energy minima, and hence reflecting a stable folding process in the system.

Thereafter, we used the insights from the free-energy landscape analysis and constructed the energy minima structures (Figure 9). These structures were in good agreement with those constructed through molecular docking analysis, reinforcing the insight about the potentially stable binding sites for Aβ peptides on ChAT protein.

The 3D cartoon representations for the top three clusters of ChAT-Aβ_40_ are given in (Figure 9K–M). The putative entrances of the catalytic tunnel on ChAT are indicated by a *red circle* (for choline/ACh entrance site; Figure 9J) and a *yellow circle* (for ACoA/-CoA entrance site; Figure 9). Intriguingly, in the cluster-1 (Aβ_40_ from the energy minima conformations) it can be appreciated that the complex seems to stabilize in a way that the N-terminal of the Aβ_40_ tends to hinder the opening of the choline/ACh entrance site of ChAT’s catalytic tunnel (*red circle* in Figure 9L), whereas the most probable cluster-0 indicated that Aβ_40_ completely covered the ACoA/-CoA entrance site on ChAT (*yellow circle* in Figure 9K) as well as the cluster-2 indicating that the Aβ_40_ covers the mouth of ACoA/-CoA entrance site on ChAT (*yellow circle* in  Figure 9M). Thus, these binding clusters analyses intuitively suggest that the Aβ_40_ binding at cluster-1 but not the cluster-0 or -2 could block the influx of the substrate, choline or ACh, and thereby the access of the enzyme to these substrates.

For ChAT-Aβ_42_ complex, the top three clusters are presented as a 3D cartoon in Figure 9O–Q, illustrating the most probable binding sites of Aβ_42_ on ChAT. Again, the position of the choline/ACh entrance site of the catalytic tunnel is highlighted by a *red circle* (in Figure 9N). The most probable cluster-0 (Figure 9O) of Aβ_42_ appeared to completely overlap the catalytic tunnel. Nonetheless, there are two additional binding sites for Aβ_42_, namely the clusters-1 and -2 (*yellow circle*; Figure 9P,Q) that completely block and are in close vicinity of the mouth of the ACoA/-CoA entrance site on the catalytic tunnel, respectively. In addition, we have compiled the simultaneous possible bindings for Aβ_42_ and Aβ_40_ as depicted in (Figure 9R). For instance, binding of Aβ_42_ at cluster-1 (Figure 9P) will most likely exclude binding of Aβ_40_ at cluster-0 (Figure 9K), or vice versa. In contrast, it is also possible that two Aβ bind simultaneously on a ChAT molecule, e.g., conditions L-Q or O-Q in (Figure 9R). The final conformations obtained at the end of the simulation of the ChAT-Aβ complexes are depicted in Appendix A. These were resembling the lowest energy conformations (shown in Figure 9) indicating that towards the end of the simulation period, the ChAT-Aβ complexes stabilized into a low energy conformation.

Furthermore, we have generated the contacts map for the ChAT-Aβ complexes (Figure 10 and Figure 11). It shows the amino acid residues of the ChAT protein that make a close contact with the Aβ peptides. This can give hints about the necessary residues that are more likely to be interacting with each other during the complex formation.

## 3. Discussion

Our previous report clearly revealed that mainly Aβ_42_ peptides potentiate the catalytic rate of ChAT by 20–30%. In contrast, Aβ_40_ potentiates ChAT activity in some analysis, but inhibits it in the others (compare Figure 1A and Figure 1C). To shed light on such discrepancies, as well as to elucidate the molecular fingerprint of the interaction between Aβ peptides and ChAT governing the CPL activity of the Aβ peptides, we conducted the current investigation using two in silico approaches, namely a molecular docking and a 100 ns-long molecular dynamics (MD) simulation analysis.

Here, we show that the molecular docking analysis identified three high probability clusters for Aβ_40_, and three for Aβ_42_. Interestingly, molecular docking identified high-probability ChAT-Aβ complex clusters for both Aβ_40_ and Aβ_42_ that had overlapping binding sites with the mouth of the Choline/ACh entrance site of the ChAT catalytic tunnel, which is expected to inhibit rather than enhance ChAT catalytic efficiency. Nonetheless, we also identified binding clusters that blocked the ACoA/-CoA entrance site.

While the overlapping binding cluster of Aβ_40_ with the choline/ACh entrance site could explain the inconsistency observed for the mode of action of Aβ_40_ peptides on ChAT in the in vitro experiments, where in some experiments Aβ_40_ inhibited, and in others it enhanced ChAT activity. Yet, such an entrance-overlapping cluster fails to explain why Aβ_42_ did not exhibit this dual mode of action. Nonetheless, there were some differences that could shed light on this difference. (1) Molecular docking and MD analyses suggested that Aβ_42_ had two additional high probability clusters that did not overlap with the mouth of the choline/ACh catalytic tunnel of ChAT (compare Figure 2 and Figure 9). (2) In addition, molecular docking analysis indicated that the Aβ_42_ had two clusters with a binding site in close vicinity of the entrance of ACoA/-CoA into the catalytic tunnel (Figure 2G,H) of which cluster-1 seems to cover the entrance of ACoA/-CoA in Lowest Energy Minima conformer obtained from MD analysis (Figure 9P). In turn, such alternative bindings could mediate a positive effect on the substrate influx into the catalytic domain upon Aβ_42_ interaction that is absent in case of Aβ_40_, and thereby on the catalytic efficiency of ChAT, in a manner that we have previously reported for the interaction of Aβ with BuChE [13].

Of note, there are two fundamental differences that exist between the catalytic modes of cholinesterases (AChE and BuChE) and ChAT. Firstly, ChAT can act reversibly, meaning that it can both produce and breakdown ACh, while ChEs only degrade ACh. Thus, ChAT always produces two products: ACh and -CoA in the forward reaction but choline and Acetyl-CoA in the reverse reaction. Secondly, the rate-limiting step in the cycle of ACh synthesis is the release of -CoA from its binding site (that is why -CoA is reported to act as an inhibitor of ChAT at concentrations higher than 50µM) [30].

Overall, the following hypothetical explanations may account for the observed changes in the catalytic activity of ChAT in the presence of Aβ peptides as reported by us (Figure 1) and the conflicting reports by others. (1) Some of the binding clusters may indicate that binding of Aβ, in particular Aβ_42_, induces a conformational change in the enzyme structure, facilitating a faster release of -CoA from its binding sites and allowing the enzyme a faster re-entry into a new ACh cycle production. (2) Other in silico clusters may indicate that Aβ might reduce the reverse reaction of ChAT i.e., prevent the conversion of ACh and -CoA to choline and Acetyl-CoA. In other words, in these binding clusters, Aβ mainly directs the forward ACh production reaction. This could occur by favoring the entrance of Choline and/or Acetyl-CoA from their respective entrances or prevent/reduce the entrance of ACh and -CoA into the catalytic domain. (3) Given that both Aβ_42_ and Aβ_40_ are expected to be present together in vivo, the most likely scenario is that both alternative mechanisms are at work simultaneously, allowing Aβ_42_ to mainly act as ChAT potentiating ligand (CPL) by implementing both alternatives, while allowing Aβ_40_ to act as a weak CPL or a blocker of Aβ_42_ and ChAT activity. The net effect would be a controlled increase in ACh production in some situations but not in others. However, based on the current data, we can neither exclude nor confirm any of these scenarios, although the overall in vitro data favors the third alternative, as indicated by the data shown in Figure 1, where a biological 10:1 ratio of Aβ_40_ to Aβ_42_ resulted in ~15% CPL activity compared to ~25% for Aβ_42_ alone.

The fourth hypothetical explanation concerns with the possibility of dual Aβ binding or exclusion of binding. For instance, the binding of Aβ_42_ at the cluster-1 site seems to be excluding the binding of Aβ_42_ to the cluster-2 site but not the cluster-0 site (Figure 2F–H). This may increase the probability of simultaneous binding of two Aβ_42_ peptides at the opposite cluster-0 and -1 site (or -0 and -2) on ChAT protein (Figure 2F and 2G or 2F and 2H). In contrast, the binding sites for cluster-0 and cluster-1 for Aβ_40_ are too close to each other, which most likely could exclude the simultaneous binding of two Aβ_40_ peptides on the hAT protein (Figure 2B,C). As Aβ_40_ binds at cluster-1, but not at cluster -0 or -2, and completely blocking the entrance of choline/ACh into the catalytic tunnel, we can expect a 50% chance for either activation or inhibition, explaining the inconsistency in the activation of ChAT by Aβ_40_ peptides.

For the docking and molecular dynamics analysis, we randomly chose one out of several conformations for Aβ_40_ and Aβ_42_ structures that were present in PDB deposition. Nonetheless, the Aβ_40_ and Aβ_42_ peptides have disordered/unstructured N- and C-terminals and can exist in several different conformations as shown by structural NMR studies [27,28]. In addition, despite the successful docking analyses by ClusPro docking server in the prediction of the binding mode of Aβ peptides to ChAT, it should be mentioned that the analyses represented a rigid-body docking, which does not allow conformational changes during protein–protein interactions. To overcome these limitations, we performed an advanced 100 ns molecular dynamics simulation on the representative clusters. These molecular dynamics analyses not only confirmed the unstructured nature of C- and N-terminals in Aβ peptides, but also provided evidence that the ChAT-Aβ complexes were stable, supporting a plausible mode of interaction between these two proteins. Overall, these in silico analyses strongly confirm a direct physical interaction between Aβ peptides and the ChAT protein, which further support our published in vitro enzyme kinetic analyses.

## 4. Materials and Methods

### 4.1. In Silico Analysis

#### 4.1.1. Molecular Docking Studies

To reveal the probable interaction sites between Aβ peptides and ChAT protein, we performed in silico molecular docking studies using a similar strategy, as described earlier for Aβ peptides vs AChE and BuChE. [13] Briefly, the X-ray crystal structure of ChAT (PDB id: 2FY3) [31], and solution structures of Aβ_42_ (PDB id: 1IYT) [27] and Aβ_40_ (PDB id: 1BA4) [28] were downloaded from the RCSB protein data bank and examined thoroughly for any artifacts. For this docking analysis, we randomly chose one out of several conformations for Aβ_40_ and Aβ_42_ structures that were present in the PDB deposition for Aβ peptides.

The structures were further prepared using the protein structure preparation tool of SYBYL-X molecular modeling suite installed on Linux based Dell Precision T7610 workstation (Intel™ Xeon™ E5-2643 CPU @ 3.3 GHz; 16GB RAM, 2 TB hard disk). The protein preparation steps involved the addition of any missing hydrogens, deletion of water molecules, and energy minimization using the Powell method [32] with a Tripos force field.

Cluspro 2.0 [33,34,35], a fully automated protein-protein docking web server was used for the docking studies. The Cluspro docking involves three main steps. As the first step, PIPER, a Fast Fourier Transform correlation-based docking method is employed to obtain a set of solutions, which is then followed by clustering of the obtained solutions based on a hierarchical pairwise root mean squared deviation (RMSD) algorithm to retain near-native conformations and discard the unstable clusters as a second step. In the third and final step, the identified clusters are evaluated for their stability using the Monte Carlo method and further refinement was performed by using the medium range optimization method. The best models are selected based on the size of the clusters and the parameters generated by the ‘Balanced’ scoring function defined by a balance between the electrostatic, hydrophobic, and van der Waals forces.

#### 4.1.2. Molecular Dynamics Simulations Analysis

Molecular dynamics study was performed using GROMACS 2020 and AMBER99SB-ILDN force field [36,37,38,39]. Briefly, the top scoring complexes were solvated in a cubic box simple point charge (SPC/E) water model keeping a distance of 1.0 nm between each side. Further, the system was solvated and ionized with Na+ and Cl^−^ ions at a concentration of 100 mM to neutralize the system. The system was then minimized using the steepest descent algorithm until the maximum force became less than 1000 kJ/mol/nm. We used position restraining during the equilibration run using constant number, volume, and temperature (NVT) and isothermal-isobaric (NPT) ensemble for 1 ns each to avoid any distortion, which might happen during the equilibration step and make the system unstable. We used V-rescale temperature coupling [40] and Parrinello–Rahman pressure coupling [41] to maintain the system at 300 K temperature and 1 bar pressure along with a coupling constant of 0.1 picosecond (ps) for temperature and 2 ps for pressure. Position restraining was used during NVT and NPT ensemble simulations and the long-range electrostatic interactions and van der Waals interactions were calculated using the Particle mesh Ewald (PME) method [42] and the cut-off for short-range van der Waals was set to 1 nm. All bonds were constrained using LINCS algorithm [43] and the time step of the simulation was set to 0.002 ps. Finally, a 100 ns productions simulation was performed, and carried out in triplicates under periodic boundary conditions. The final production simulation trajectory was analyzed quantitatively by calculating root means square deviation (RMSD), root mean square fluctuation (RMSF), radius of gyration (Rg), solvent accessible surface area (sasa) and secondary structure analysis using rmsd, rmsf, gyrate, sasa and do_dssp functionalities, respectively. H-bond functionality was used to analyze the hydrogen bond formed between ChAT and Aβ peptides and also the hydrogen bond distance throughout the simulation time.

Protein molecules have entrenched movements that are mainly due to their complex association within their atomic motions, which are of utmost importance in regular conventional functioning of the protein. These internal motions of a protein molecules are difficult to interpret. Thus, PCA analysis was also carried out with the help of modules such as gmx covar and gmx anaeig available on GROMACS. The motion in MD trajectories was analyzed, which gives an insight into the crucial motions of the simulated system [44,45]. PCA was carried out for the ChAT-Aβ peptides complex using the gmx covar and gmx anaeig in two steps: (a) constructing the covariance matrix using the Backbone atoms and (b) diagonalization of the covariance matrix of the atomic coordinates. The relative motion of the complex system was obtained as the projection of the first two eigenvectors, which accounted for containing the maximum motion of the system. The eigenvectors are a representation of the direction of the motion, whereas eigenvalues represent the magnitude of motion along with the direction. On these projections, we can see the clusters of stable states. The overall flexibility of the ChAT-Aβ peptides complex was calculated by the trace of the diagonalized covariance matrix of the backbone atomic positional fluctuations. Moreover, the free-energy landscape was analyzed using the gmx sham module available on GROMACS, and the contact maps for the amino acid residue interactions between ChAT protein and Aβ peptides were evaluated to gain insight into the mechanism of how Aβ peptides interact with the ChAT protein.

## 5. Conclusions

In light of the failure of numerous Aβ-directed clinical drug trials in the AD field, it is now of utmost important that we should take a step back and focus more on understanding the native function of Aβ peptides and their involvement in AD-related cholinergic dysfunction. Here, we performed advanced in silico analyses (molecular docking and molecular simulation studies) to further understand the observed in vitro direct interacting mechanism between the key-cholinergic enzyme, ChAT and Aβ peptides [26]. The molecular docking analyses suggested three most probable binding clusters on ChAT for Aβ_40_ and Aβ_42_. Interestingly, one binding cluster of each Aβ_40_ and Aβ_42_ overlapped the choline/ACh entrance site on the ChAT catalytic tunnel, which is expected to inhibit rather than enhance ChAT catalytic efficiency. The results of the molecular dynamics simulation studies clearly demonstrated that the ChAT-Aβ complexes were energetically stable despite the unstructured nature of C- and N-terminals in Aβ peptides and formed anticipated hydrogen bonds during simulated interaction studies. The present in silico analyses hence further reinforce our published in vitro ChAT-Aβ interaction study. Overall, our findings provide new information and shed light on the potential unexplored complex molecular crosstalk between Aβ and the enzymatic machinery involved in maintaining cellular, synaptic, and extra-synaptic ACh homeostasis [13,26]. We expect these insights to open new avenues for both a proper understanding of the pathophysiological role of Aβ peptides related specifically to the cholinergic dysfunction in AD, as well as for developing a new class of cholinergic enhancing drugs that mimic the CPL activity of Aβ_42_ peptides for the treatment of the cholinergic deficit seen in AD-related dementias and the neuromotor disorders such as ALS.

## Figures and Tables

**Figure 1 ijms-23-06073-f001:**
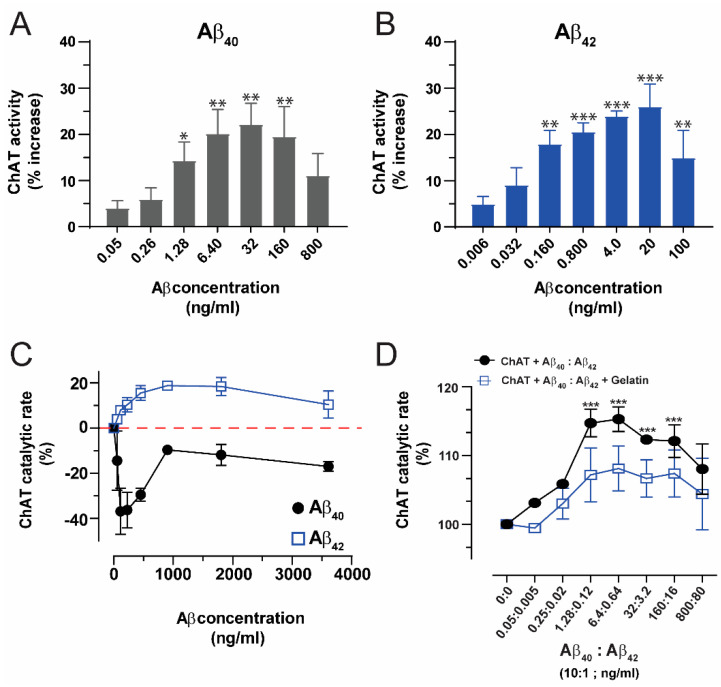
Soluble Aβ peptides specifically and allosterically activate ChAT at physiological concentrations via direct interaction (adapted and reproduced from Kumar et al. [26], under the terms of the CC BY license). ChAT was incubated with Aβ peptides at physiological concentration ranges and changes in its activity were monitored using the kinetic ChAT fluorometric assay. (**A**) The percent changes in the rate of ChAT (ΔRFU/hr) by Aβ_40_ at the specified concentration range of 0 to 800 ng/mL (**B**) The corresponding percent of changes in ChAT activity by Aβ_42_, at the specified concentration range of 0 to 100 ng/mL. The activity of the control sample (ChAT with no Aβ) was considered as 100 % to calculate the percentage increase in ChAT activity in the presence of different Aβ concentrations. The values are shown as mean ± SEM of ≥3 independent experiments. (**C**) The effect of high concentrations (non-physiological range) of Aβ peptides on ChAT activity was also monitored. ChAT was incubated with different concentrations of Aβ_40_ and Aβ_42_ peptides, ranging from 0 to 3800 ng/mL in 96-well plates. The values are shown as mean ± SEM of two independent experiments. For Aβ_40_ (**A**), the symbols * and ** represents *p*-values < 0.005 and < 0.0003, respectively, relative to the control (no Aβ_42_). The comparisons are based on one-way ANOVA analysis and Fisher’s LSD post-hoc test. (**D**) Both Aβ_40_ and Aβ_42_ are present simultaneously in biological fluids at a 10:1 ratio. Therefore, we measured ChAT activity in the presence of various concentrations of mixtures of Aβ peptides to somewhat mimic the in vivo conditions. Graph shows the percent changes in ChAT activity at the specified Aβ mixture concentrations with or without gelatin (which mimics a high protein concentration microenvironment, as expected physiologically in biological fluids). The activity of the control sample (ChAT with no Aβ) was considered 100%. The values are shown as mean ± SEM of ≥ 2 independent experiments. *** *p* < 0.0001, relative to the control (no Aβ), based on one-way ANOVA analysis and Fisher’s LSD post-hoc test.

**Figure 2 ijms-23-06073-f002:**
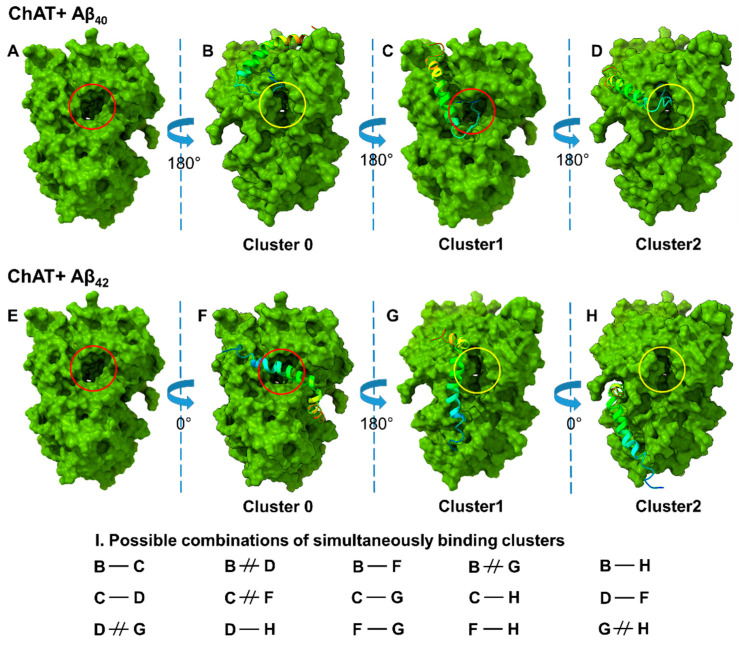
In silico molecular docking analyses reveal the potential binding sites for Aβ peptides on ChAT. Molecular docking for ChAT and Aβ peptides was performed using Cluspro web server as described in the method section. (**A**,**E**) show the 3D representation of ChAT protein, where the choline-entrance of the catalytic tunnel is indicated by *red circle*. (**B**–**D**) show the most probable binding-sites for clusters-0, -1 and -2 of Aβ_40_ peptides, respectively. The *yellow circle* highlights the Acetyl-CoA-entrance of the catalytic tunnel. A comparison of (**B**–**D**) indicates that the binding of Aβ_40_ at cluster-1, but not cluster-0 or -2, will most likely block the entrance of choline into the catalytic tunnel of ChAT, whereas Aβ_40_ binding in cluster-2 will interfere with the Acetyl-CoA-entrance site. (**F**–**H**) illustrate the most probable binding-site clusters -0, -1, and -2 for Aβ_42_ peptide, respectively. Similar to Aβ_40_, one out of the top three clusters of Aβ_42_ (cluster-0) also overlaps the choline-entrance site of the catalytic tunnel (**F**). Theoretically, it is possible that two Aβ peptides can simultaneously bind to one ChAT molecule. The table (**I**) shows all such possible/excluding combinations. For example, the cluster-0 (**B**) will most likely exclude simultaneous binding of another peptide at cluster-2 (**D**) or vice versa. This is denoted as B ≠ D. In contrast, it is possible for two Aβ_42_ peptides to bind simultaneously to ChAT, namely at cluster -0 and -1 (**F**–**G**), or at cluster-0 and -2 (**F**–**H**).

**Figure 3 ijms-23-06073-f003:**
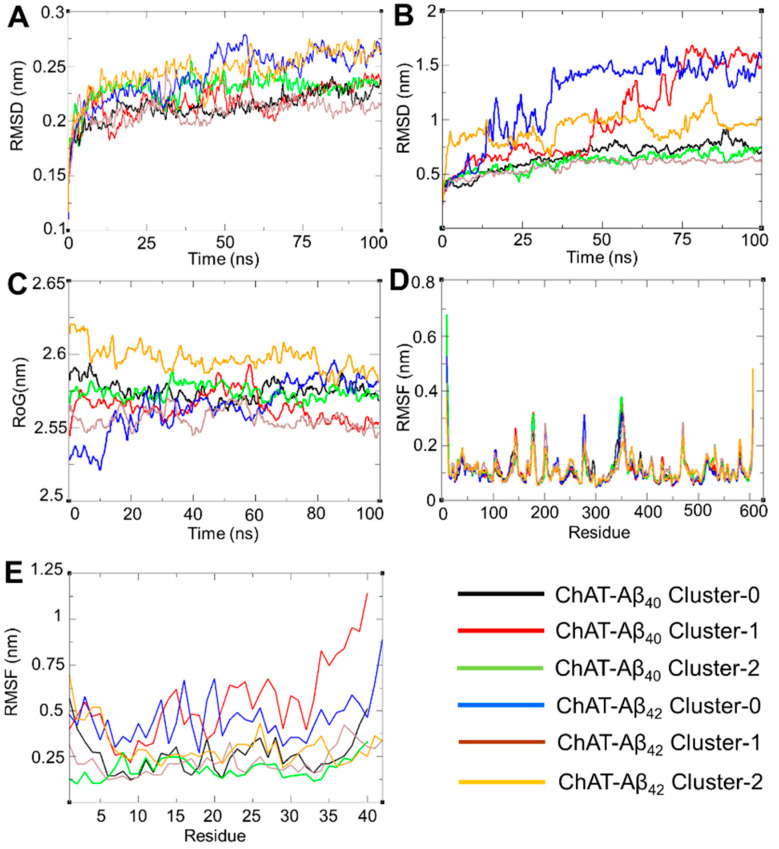
Molecular dynamics simulation analyses support the potential interaction and stability between Aβ peptides and ChAT. (**A**) represents root mean squared deviation (RMSD) plots as a function of simulation period for the ChAT protein. (**B**) shows the root mean squared deviation (RMSD) plots for Aβ peptides. (**C**) illustrates the radius of gyration (RoG) plots for the complexes. (**D**,**E**) show root mean squared fluctuation (RMSF) for ChAT and Aβ amino acid residues, respectively. The clusters are represented by the given color scheme. All of the graphs were plotted with the QtGrace software package.

**Figure 4 ijms-23-06073-f004:**
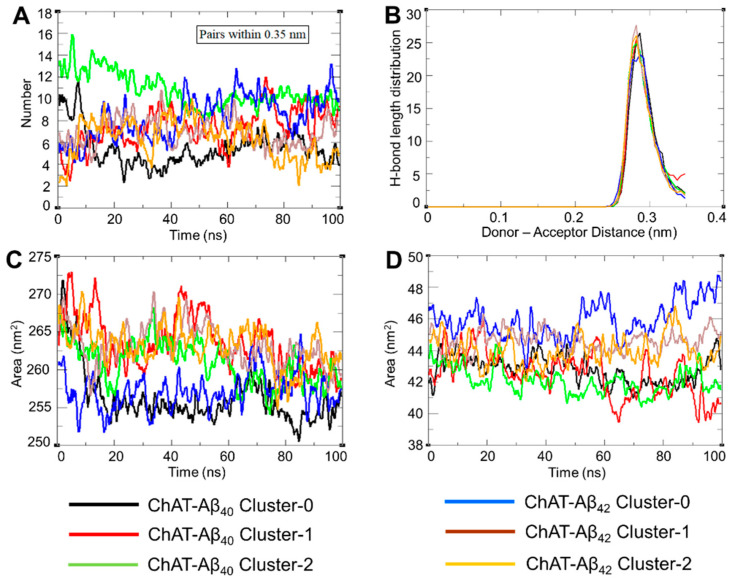
The ChAT-Aβ complex formation and stabilization based on hydrogen bonding landscape analysis. H-bonding plays a central role during protein interactions; thereby (**A**) shows the average number of hydrogen bonds formed between ChAT-Aβ peptides during 100 ns simulation trajectory. (**B**) illustrates the average hydrogen-bond distance around the cut-off for donor–acceptor distance that was set at 0.35 nm. (**C**,**D**) denote the change in total solvent accessible surface area (SASA) of the ChAT protein and the total solvent accessible surface area of the Aβ peptides, respectively, corresponding to the simulation time-period of the complex system. The clusters are represented by the given color scheme.

**Figure 5 ijms-23-06073-f005:**
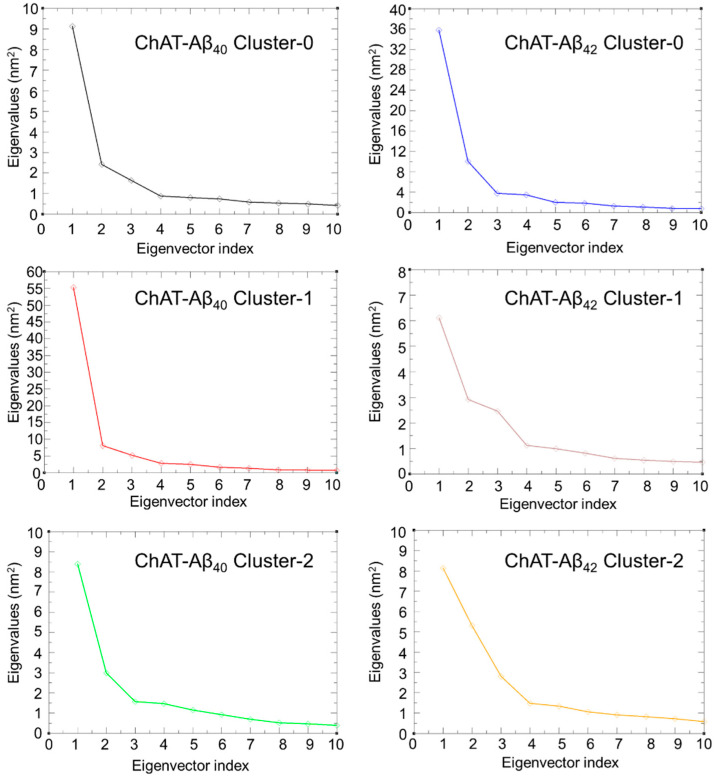
This represents the eigenvalues of each principal component for all the ChAT-Aβ peptides complex simulated systems. Principal component (PC) analysis tries to find combinations of features that leads to the maximum separation between the data points. The analysis suggests that the first three PC eigenvector accounts for most of the motions in the system, while from the fourth PC eigenvector the contribution become less important. We used the first two PCs to generate the cluster projections by plotting it on x and y axis, respectively, to explain the data points.

**Figure 6 ijms-23-06073-f006:**
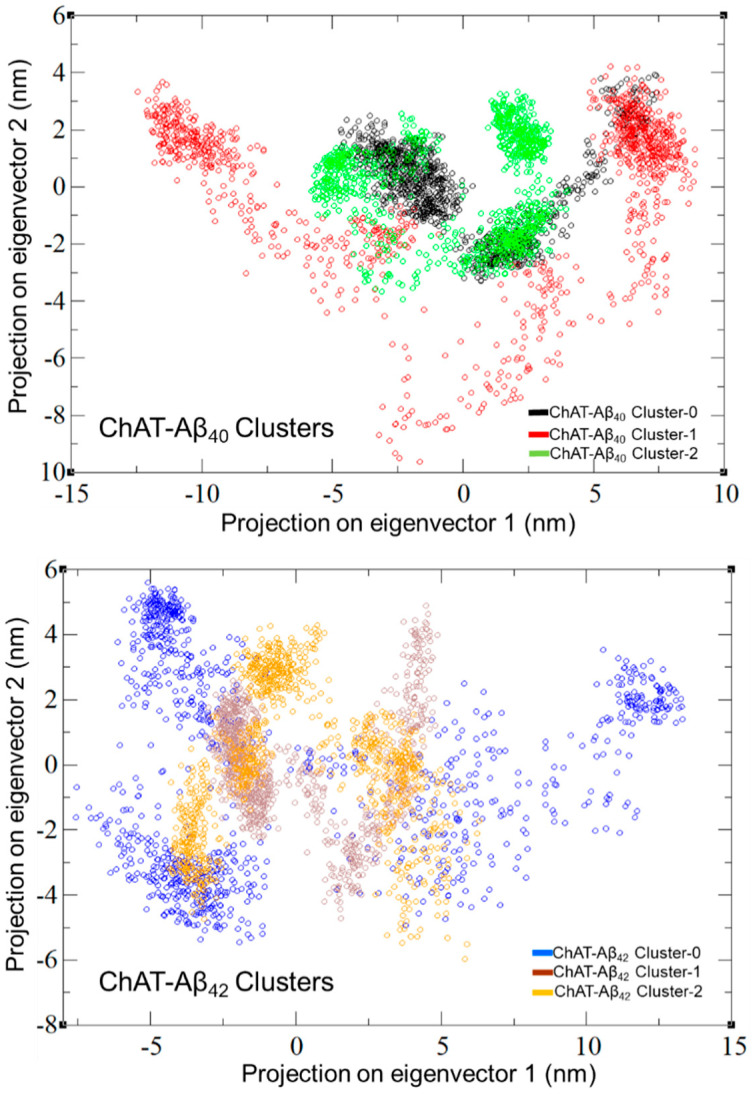
Representation of the projection of the first two principal components for the ChAT-Aβ complex system. Each dots represent a specific conformational state of the ChAT protein in the complex system with Aβ. Congregated clusters of the dots indicate that the protein did not go through any major conformational changes, whereas the widespread dots indicate more flexibility of the protein in the complex and, hence, some major conformational changes that might take place. Overall, the graphs illustrate the close concentrated distribution of the conformational sampling for the majority of the ChAT-Aβ complexes clusters except for the ChAT-Aβ_40_ cluster-1 and ChAT-Aβ_42_ cluster-0, which exhibit widespread distribution of conformational sampling.

**Figure 7 ijms-23-06073-f007:**
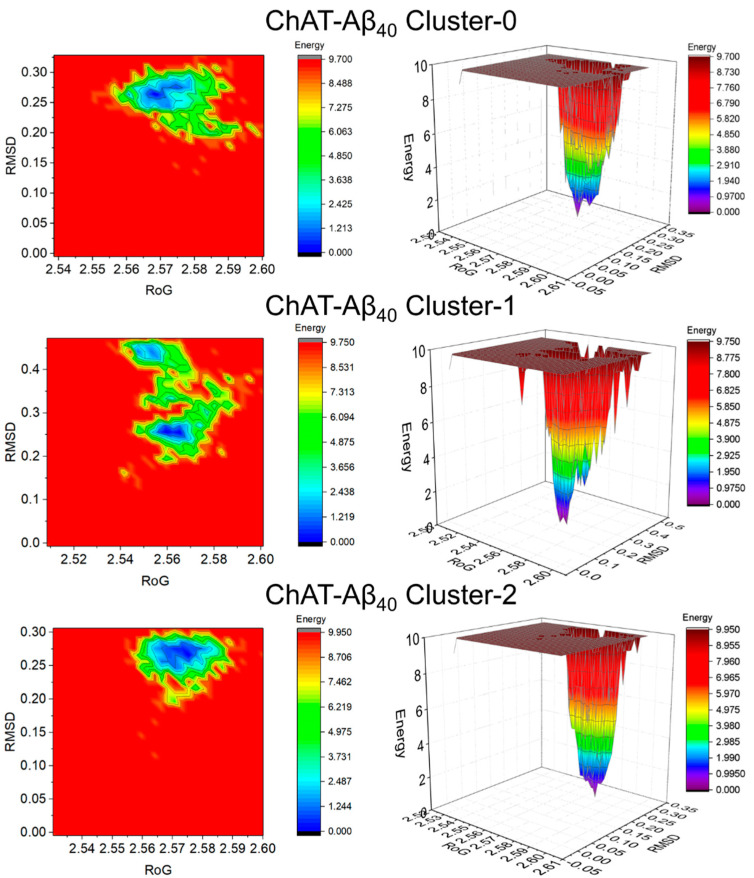
The 2D and 3D free-energy landscape diagrams as a function of RMSD and RoG as the two reaction coordinates of ChAT-Aβ_40_ clusters. The free energy is given in kJ/mol and indicated by color. The folding funnel of the complex system into a single narrow funnel indicate a stable folding process. The 2D contour plots and the 3D projections were plotted using OriginPro 2017 software (OriginLab Inc., Northampton, MA, USA). RMSD: Root mean square deviation, RoG: Radius of gyration.

**Figure 8 ijms-23-06073-f008:**
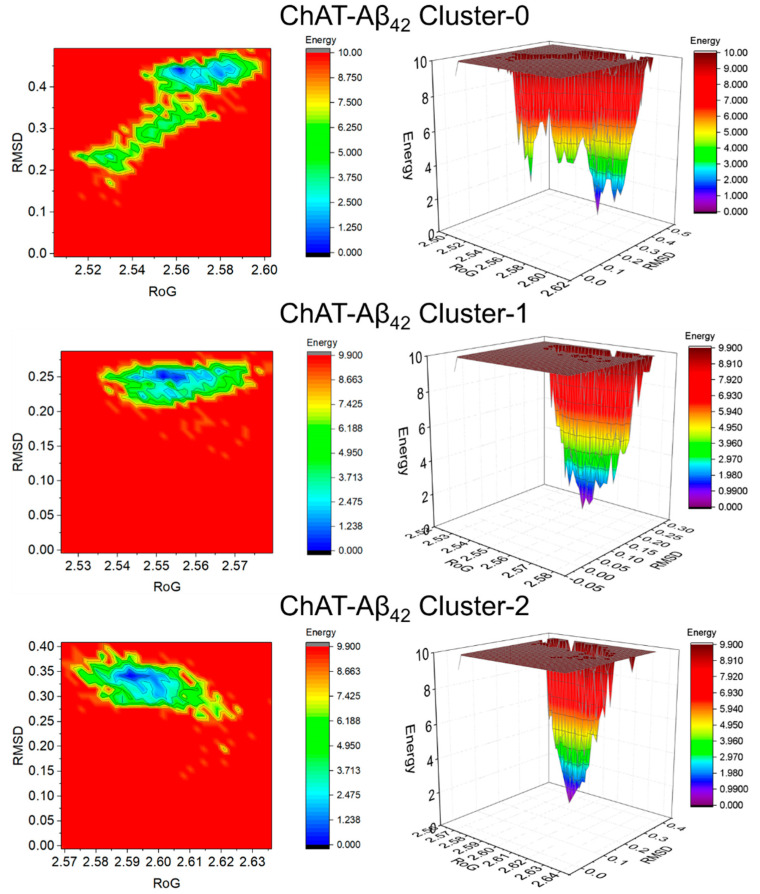
The 2D and 3D free-energy landscape diagrams as a function of RMSD and RoG as the two reaction coordinates of ChAT-Aβ_42_ clusters. The folding funnel of the complex system forms a single narrow funnel which indicates a stable folding process. The 2D contour plots and the 3D projections were plotted using OriginPro 2017 software (OriginLab Inc., Northampton, MA, USA). The free energy is given in kJ/mol and indicated by color. RMSD: Root mean square deviation, RoG: Radius of gyration.

**Figure 9 ijms-23-06073-f009:**
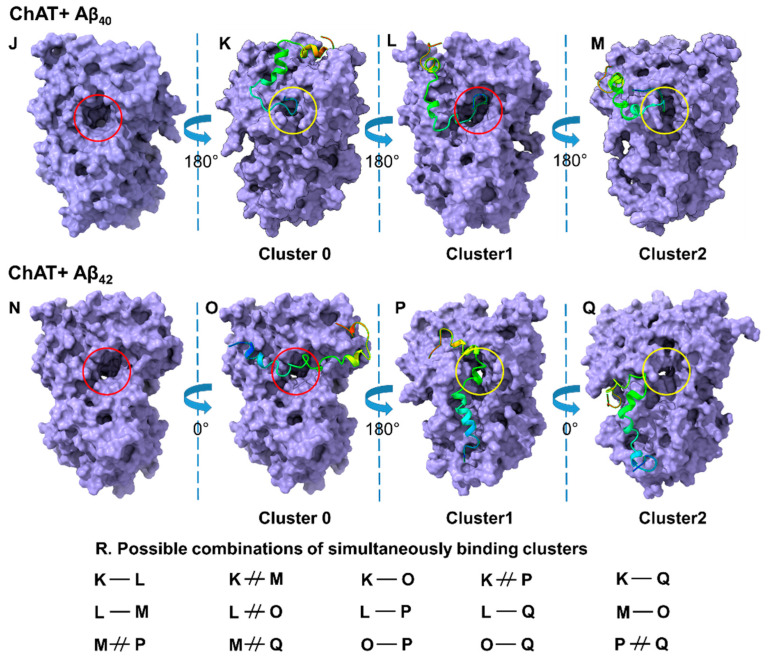
The most stable energy minima conformers of ChAT-Aβ complexes. The outlined structures represent the Lowest Energy Minima conformers, i.e., the energetically favored complexes, constructed based on the free-energy landscape analysis as a function of root mean square deviation (RMSD) and radius of gyration (RoG) of ChAT-Aβ clusters. (**J**,**N**) show the 3D representation of ChAT protein, where the choline entrance of the catalytic tunnel is indicated by a *red circle*. (**K**–**M**) show the Lowest Energy Minima conformers of the most probable binding-site clusters-0, -1 and -2 of Aβ_40_ peptides, respectively. The *yellow circle* highlights the Acetyl-CoA-entrance of the catalytic tunnel. A comparison of (**K**–**M**) indicates that the binding of Aβ_40_ at cluster-1, but not cluster-0 or -2, will most likely block the entrance of choline into the catalytic tunnel of ChAT. (**O**–**Q**) illustrate the Lowest Energy Minima conformers of the most probable binding-site clusters-0, -1, and -2 for Aβ_42_ peptide, respectively. Similar to Aβ_40_, one out of the top three clusters of Aβ_42_ (cluster-0) also overlaps the choline-entrance site of the catalytic tunnel (**O**). The table (**R**) shows all possible/excluding combinations if two Aβ peptides bind to one ChAT molecule simultaneously. For example, the cluster-0 (**K**) will most likely exclude simultaneous binding of another peptide at cluster-2 (**M**) or vice versa. This is denoted as K ≠ M. In contrast, it is possible for two Aβ_42_ peptides to bind simultaneously to ChAT, namely at cluster -0 and -1 (**O**,**P**), or at cluster-0 and -2 (**O**–**Q**).

**Figure 10 ijms-23-06073-f010:**
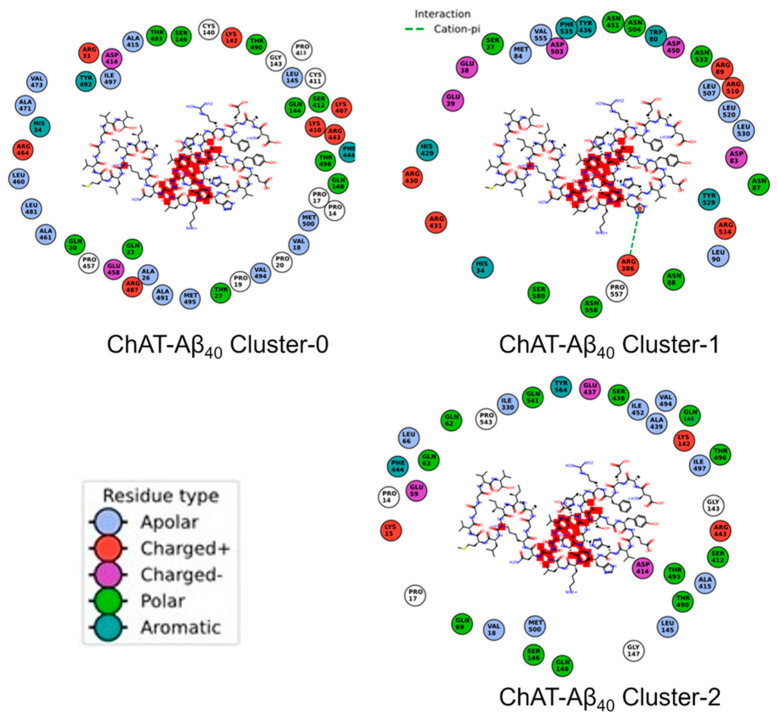
Contact maps between the ChAT protein and Aβ_40_ peptides in the complexes. The amino acid residue types are given with the color codes as blue (Apolar), red (‘+’ charged), purple (‘-’ charged), light green (Polar), dark green (Aromatic).

**Figure 11 ijms-23-06073-f011:**
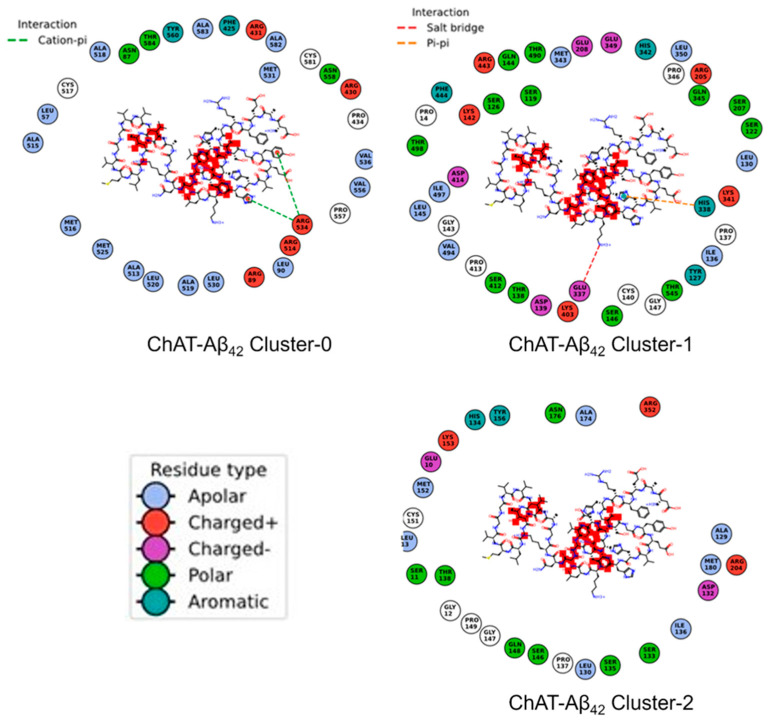
Contact maps between the ChAT protein and Aβ_42_ peptides in the complexes. The amino acid residue types are given with the color codes as blue (Apolar), red (‘+’ charged), purple (‘-’ charged), light green (Polar), dark green (Aromatic).

**Table 1 ijms-23-06073-t001:** Probabilistic binding clusters of Aβ_40_ and Aβ_42_ on ChAT.

Cluster	Representative Complex	ChAT-Aβ_40_		ChAT-Aβ_42_	
		Cluster Size	Weighted Score (Balanced)	Cluster Size	Weighted Score (Balanced)
**0**	Center	88	−920.6	134	−973.8
	Lowest Energy	88	−989.5	134	−999.6
**1**	Center	75	−850.2	94	−926.2
	Lowest Energy	75	−1011.0	94	−1111.3
**2**	Center	51	−966.0	84	−884.8
	Lowest Energy	51	−966.0	84	−991.3
**3**	Center	46	−878.0	66	−851.2
	Lowest Energy	46	−975.8	66	−875.5
**4**	Center	45	−826.8	51	−858.4
	Lowest Energy	45	−905.4	51	−858.4
**5**	Center	41	−909.3	46	−851.0
	Lowest Energy	41	−951.7	46	−851.0
**6**	Center	38	−815.5	23	−883.5
	Lowest Energy	38	−936.2	23	−883.5
**7**	Center	36	−824.2	22	−911.2
	Lowest Energy	36	−905.1	22	−911.2
**8**	Center	35	−976.1	20	−822.4
	Lowest Energy	35	−976.1	20	−888.7
**9**	Center	34	−928.1	20	−920.8
	Lowest Energy	34	−949.1	20	−920.8
**10**	Center	27	−833.2	18	−813.1
	Lowest Energy	27	−920.9	18	−932.6

ChAT = choline acetyltransferase, Aβ = Amyloid-β peptide.

**Table 2 ijms-23-06073-t002:** Average secondary structure across the trajectory clusters of Aβ_40_ on ChAT.

% Secondary Structure	ChAT-Aβ_40_ Cluster-0		ChAT-Aβ_40_ Cluster-1		ChAT-Aβ_40_ Cluster-2	
	ChAT	Aβ_40_	ChAT	Aβ_40_	ChAT	Aβ_40_
**Coil**	21	26	22	34	21	30
**Beta sheet**	14	0	14	0	15	0
**Beta bridge**	0	0	1	0	0	0
**Bend**	10	13	10	17	9	14
**Turn**	11	32	11	20	11	18
**A-helix**	40	20	40	17	40	25
**5-helix and 3-helix**	4	10	3	12	3	13

**Table 3 ijms-23-06073-t003:** Average secondary structure across the trajectory clusters of Aβ_42_ on ChAT.

% Secondary Structure	ChAT-Aβ_42_ Cluster-0		ChAT-Aβ_42_ Cluster-1		ChAT-Aβ_42_ Cluster-2	
	ChAT	Aβ_42_	ChAT	Aβ_42_	ChAT	Aβ_42_
**Coil**	21	21	21	18	22	21
**Beta sheet**	14	0	14	0	14	0
**Beta bridge**	1	0	1	0	0	0
**Bend**	10	14	10	8	10	11
**Turn**	11	32	12	24	10	26
**A-helix**	40	21	40	42	41	23
**5-helix and 3- helix**	3	13	2	7	3	19

## Data Availability

The trajectories and topology files for all six clusters obtained after molecular dynamics simulation are archived on Zenodo, doi: 10.5281/zenodo.5912821.

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
