# Peer review of "Allosteric Binding Sites of Aβ Peptides on the Acetylcholine Synthesizing Enzyme ChAT as Deduced by In Silico Molecular Modeling"

_ijms, 2022, doi:10.3390/ijms23116073_

Round 1
Reviewer 1 Report
Authors significantly improve the manuscript and it can be accepted in present form. For readers sake I recommend Authors to merge the SI Figures into one pdf file and add Figure captions.
Author Response
Please see the file "Response to Reviewers"!

Reviewer 2 Report
The paper entitled “Allosteric binding sites of A peptides on the acetylcholine 2 synthesizing enzyme, ChAT as deduced by in silico molecular 3 modeling” has a crucial topic but it needs a revision since the rationale and the scope of this study are almost unclear.
Introduction: no references on the druggability of the PPI indicated, how this protein complex can be utilized in terms of therapeutic strategies?
Results: It is not possible to reproduce a figure from another publication as a result. The scientific assumptions of a study should be clearly delineated in the introduction section, thus considerations related to the actual figure 1 should be shifted into the introduction and related data could be presented as a table putting emphasis on the points that will be investigated during the study.
Docking analyses are rigorously carried out even if the authors hypothesized a possible “Aβ42 induces a conformational change in the enzyme structure facilitating a faster release of -CoA from its binding sites”, what about the experimental proof of these considerations? Have the authors an idea through structural experimental assays?
In addition, what is the experimental affinity between proteins? What about the kinetic?
Docking studies have been carried out on the proteins as monomers, what is the effect of this PPI on the oligomerization/aggregation of beta amyloid?
Author Response
Please see the file "Response to Reviewer"!

Round 2
Reviewer 2 Report
The authors have partially satisfied all my concerns except for the KD values of interaction that should be reported in molarity and not as mass/volume.
In addition, the fact that a conformational variation is a hypothesis should be underlined, explicitly.
Author Response
- The authors have partially satisfied all my concerns except for the KD values of interaction that should be reported in molarity and not as mass/volume.
Authors’ response: Although we understand your concern, we would like to mention that we provided the values in pg/mL to simplify the matter for the readers. The Aβ concentrations in the literatures is invariably given in pg/mL so the readers in the AD field easily can compare the values with what is expected about Aβ concentrations in e.g., human CSF in control and AD cases.
Action: Nonetheless, we changed the text and additionally included the values in molarity based on the theoretical Mw provided by the manufacturer, r-peptide (4,329 Da for Aβ40 and 4,514 Da for Aβ42) as follows:
In line 85-87: “In the primary paper of our study, we have reported that Aβ40 exhibited CPL activity with an EC50 value of 800 pg/mL (or 185 pM) while Aβ42 peptides showed 10 folds higher CPL potency as deduced by an EC50 value of 68 pg/mL (or 15 pM).”
- In addition, the fact that a conformational variation is a hypothesis should be underlined, explicitly.
Authors’ response: We agree, although we believe that we were already quite specific about this issue since we used “following theoretical possibilities”.
Action: Nonetheless, we changed the text as follows
In line 484: “Overall, the following hypothetical explanations may….”
In line 503: “The fourth hypothetical explanation concerns with….”
This manuscript is a resubmission of an earlier submission. The following is a list of the peer review reports and author responses from that submission.
Round 1
Reviewer 1 Report
Authors used three starting point for their MD simulations which is correct point. However, there is only one trajectory per starting point. At least 3 trajectories per starting points need to be performed.
Final structures after simulations needs to be added to SI.
Contact maps between ChAT and Ab after simulation needs to be computed.
Reviewer 2 Report
Allosteric binding sites of Aβ peptides on the acetylcholine synthesizing enzyme, ChAT as deduced by in silico molecular modeling
Manuscript ID: ijms-1599766
Comments and Suggestions
The authors explain the interaction of Aβ peptides (Aβ40 and Aβ41) with choline acetyltransferase (ChAT) and outlined different binding sites of Aβ peptides with ChAT protein using Docking and Molecular Dynamics (MD) techniques.
1). In line 127, authors state justification of predicted models in different energy terms. Explanation of “well-justified in terms of electrostatic energy, van der Waals energy” is not explained clearly.
2). Line 154: A punctuation mark (.) expected after, -entrance site.
3). Line 227: Cut-off distance for H-bond is mentioned. Cut-off angle for hydrogen bond is expected.
4). Line 237: Authors state that SASA profile of ChAT-Aβ complex is consistent with its radius of gyration. The type of “consistency” needs to be explained.
5). Figure 2 and Figure 9 show the possible binding site of Aβ peptides with ChAT. It can be seen that Aβ binds near the catalytic tunnel of ChAT. In ChAT- Aβ40 complex, the terminal end of Aβ40 which is highly flexible as indicated by the RMSF plot, occupies the entrance of catalytic tunnel. Does this terminal end stay near to the entrance of catalytic tunnel during the entire simulation to block the entrance?
6). Authors discuss H-bond numbers in the interaction. Are there any specific residues of Aβ peptides (Aβ40 and Aβ41) and ChAT that contribute more to the H-bond formation?
Round 2
Reviewer 1 Report
Authors did not perform simulations. They justify with (incorrect resources calculation with double counting of number of system and length of simulation) The length should be 100ns as previously only the number of system simulation should be larger. With 6 low cost graphical cards (GTX 980) the results for AMBER force field (which would be sufficient for the sake of this publication) would require 2 days.
https://ambermd.org/gpus16/benchmarks.htm
